

# Advancing healthcare through multimodal data fusion: a comprehensive review of techniques and applications

Jing Ru Teoh[1], Jian Dong[2], Xiaowei Zuo[3], Khin Wee Lai[1], Khairunnisa Hasikin[1,4] and Xiang Wu[1,5]

[1] Department of Biomedical Engineering, University of Malaya, Kuala Lumpur, Malaysia
[2] China Electronics Standardization Institute, Beijing, China
[3] Department of Psychiatry, The Affiliated Xuzhou Oriental Hospital of Xuzhou Medical University, Xuzhou, Jiangsu, China
[4] Faculty of Engineering, Centre of Intelligent Systems for Emerging Technology (CISET), Kuala Lumpur, Malaysia
[5] Institute of Medical Information Security, Xuzhou, Jiangsu, China

Corresponding authors
Jian Dong, dongjian@cesi.cn
Khin Wee Lai, lai.khinwee@um.edu.my

## ABSTRACT

With the increasing availability of diverse healthcare data sources, such as medical images and electronic health records, there is a growing need to effectively integrate and fuse this multimodal data for comprehensive analysis and decision-making. However, despite its potential, multimodal data fusion in healthcare remains limited. This review paper provides an overview of existing literature on multimodal data fusion in healthcare, covering 69 relevant works published between 2018 and 2024. It focuses on methodologies that integrate different data types to enhance medical analysis, including techniques for integrating medical images with structured and unstructured data, combining multiple image modalities, and other features. Additionally, the paper reviews various approaches to multimodal data fusion, such as early, intermediate, and late fusion methods, and examines the challenges and limitations associated with these techniques. The potential benefits and applications of multimodal data fusion in various diseases are highlighted, illustrating specific strategies employed in healthcare artificial intelligence (AI) model development. This research synthesizes existing information to facilitate progress in using multimodal data for improved medical diagnosis and treatment planning.

# INTRODUCTION

Automation in healthcare processes through the application of artificial intelligence (AI) has the capacity to bring transformative changes. However, in most AI applications, the predominance reliance on unimodal data such as computed tomography (CT) scans, magnetic resonance imaging (MRI), X-rays images *etc.* presents unique challenges in modern healthcare applications. These models frequently fail to incorporate crucial

complementary data sources and various modalities, which limits their capacity to provide comprehensive insights (*El-Sappagh et al., 2020*; *Moshawrab et al., 2023*).

Healthcare AI applications are predominantly dominated by single-task models that rely on singular data types, lacking comprehensive clinical context. This contrasts with the holistic methods favored by clinicians and signifies a missed opportunity. Neglecting to utilize multimodal systems, which integrate multiple data modalities and interdependent tasks, hinders treatment efficacy and diagnostic accuracy (*Acosta et al., 2022*; *El-Sappagh et al., 2020*). Despite their potential for more accurate and comprehensive outcomes, these systems remain limited in implementation. Embracing multimodal data integration offers a promising solution, paving the way for AI-driven healthcare capable of nuanced diagnoses, precise prognostic evaluations, and tailored treatment plans.

The limitations are particularly significant in the fields of radiological image interpretation and clinical decision support systems. Radiologists facing overwhelming image interpretations encounter increased fatigue and higher error rates. Meanwhile, despite proficiency in image analysis, automated systems often struggle to integrate critical clinical context, akin to human physicians' meticulous approach (*Huang et al., 2020a*). The importance of integration becomes evident in medical imaging interpretations, where the fusion of heterogeneous data sources including imaging findings, patient demographics, clinical history, and risk factor information is essential.

Furthermore, integrating diverse data modalities in biomedical research has proven beneficial in understanding complex diseases like cancer. For instance, fusing genomic data with histopathological images provides crucial insights into cancer heterogeneity, aiding tailored therapies and improving predictions (*M'Sabah et al., 2021*; *Stahlschmidt, Ulfenborg & Synnergren, 2022*). The convergence of various data types consistently demonstrates improved diagnostic accuracy across multiple medical imaging tasks (*Huang et al., 2020b*; *Mammoottil et al., 2022*; *Sun et al., 2023*). The motivation behind utilizing multimodal data in healthcare is its demonstrated ability to substantially enhance diagnostic accuracy, enable personalized treatments, optimize resource allocation, and improve overall healthcare delivery. These advancements promise transformative shifts towards comprehensive healthcare solutions catering to individual patient needs.

In this paper, the terminology 'data fusion' refers to the technique of integrating multiple data modalities, while 'multimodal data' refers to the combined dataset resulting from this integration. In this study, a fusion of medical healthcare data to form multimodal data using different types of fusion techniques is conducted to collect and synthesize the available literature to establish a foundation for future research. We aim to find all relevant information regarding the fusion techniques of multimodal data and different types of data combinations. In addition, most of the review papers focused on fusion techniques and strategies and surveyed recent trends and advances. In this study, the contributions of this paper are as follows:

1. Our focus extended to analyzing various data fusion techniques in healthcare AI model development. By examining each fusion method, we provided comprehensive insights into the healthcare data fusion landscape, offering valuable guidance for researchers and practitioners.

2. We focused on various multimodal data fusion techniques, including the integration of medical images with structured data, unstructured data, multiple image modalities, and other features. By exploring these techniques, we clarified the strategies applied in healthcare AI model development.

3. We highlighted the applications of multimodal data in various diseases to gain a clear view of the fusion techniques used for specific types of diseases.

In this paper, we have carefully identified and reviewed 69 related works published between 2018 and 2024 that employed data fusion techniques in combining multiple modalities of healthcare data. This paper also provides the links and websites of the public datasets that are normally used by researchers in particular diseases. The paper is organized in the following structure: the Methodology section provides the article selection of this study; the Results section presents data fusion techniques of multimodal data, reviews the papers with related works on data fusion techniques, and discusses different multimodal data in various diseases; and the Discussion section offers a comprehensive discussion of the proposed framework and future works.

## METHODOLOGY

### Article selection

In this review paper, we systematically selected relevant studies based on specific inclusion and exclusion criteria to ensure comprehensive coverage and quality, as shown in Fig. 1. The inclusion criteria encompassed papers published from 2018 onwards in the Web of Science (WOS) database, illustrated in Fig. 2. The year 2018 was chosen because it marks the widespread introduction of multimodal data fusion in the healthcare sector. We utilized WOS as our primary resource for finding articles due to its advantages: it is recognized as a reliable and comprehensive database, containing high-quality scholarly journals from various fields, and it implements rigorous quality control measures, such as peer review and citation analysis, to ensure the reliability and credibility of the included literature.

In addition, the following research questions were developed formulated in aiding the process of developing this review:

1. What are the techniques used to fuse multimodal data?
2. Which data can be used to form multimodal data?
3. What are the applications of multimodal data fusion?
4. How does multimodal data benefit the healthcare sector?
5. What types of AI models mostly developed by the researchers using data fusion?
6. What are the challenges and possible solutions towards AI model development using multimodal data fusion?

### Search string

To find relevant articles regarding multimodal data fusion, we used the following search string:

*("Multimodal Fusion" OR "Multimodal Data Fusion") AND ("Medical") AND ("Disease") AND ("Data Fusion") AND ("Early Fusion") AND ("Intermediate Fusion" OR "Joint Fusion") AND ("Late Fusion")*

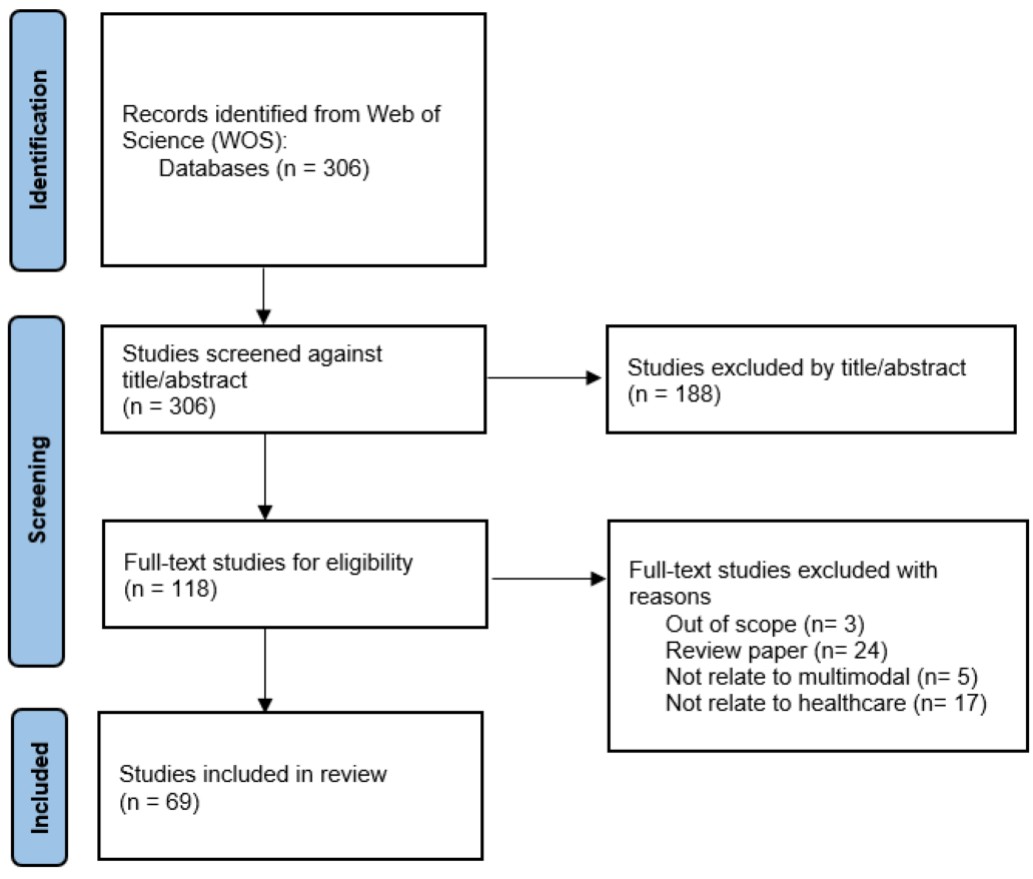

**Figure 1   The PRISMA flowchart of article selection process.**

The keywords were chosen to specifically denote the integration of multiple data modalities and ensure the search is confined to the medical field, the primary focus of our study. These keywords help capture studies related to various diseases, making the results pertinent to understanding how multimodal data fusion can aid in disease diagnosis, prognosis, or treatment.

## Study selection criteria

We identified relevant studies using specific selection criteria, considering only articles directly related to the medical field or healthcare sector and employing multimodal data fusion techniques. The title and abstract of each paper were assessed for relevance to our objective of reviewing multimodal data fusion in healthcare. To maintain rigor, we excluded case studies, news items, review papers, and non-English articles. Only articles with full-text access were included to ensure thorough examination and analysis. By adhering to these criteria, we aimed to select high-quality and pertinent studies for our review.

We used VOSviewer to gain insights into the academic landscape of our field. This software visualizes and analyzes bibliometric data. We created co-authorship networks to understand researcher collaboration and analyzed citation networks to identify influential

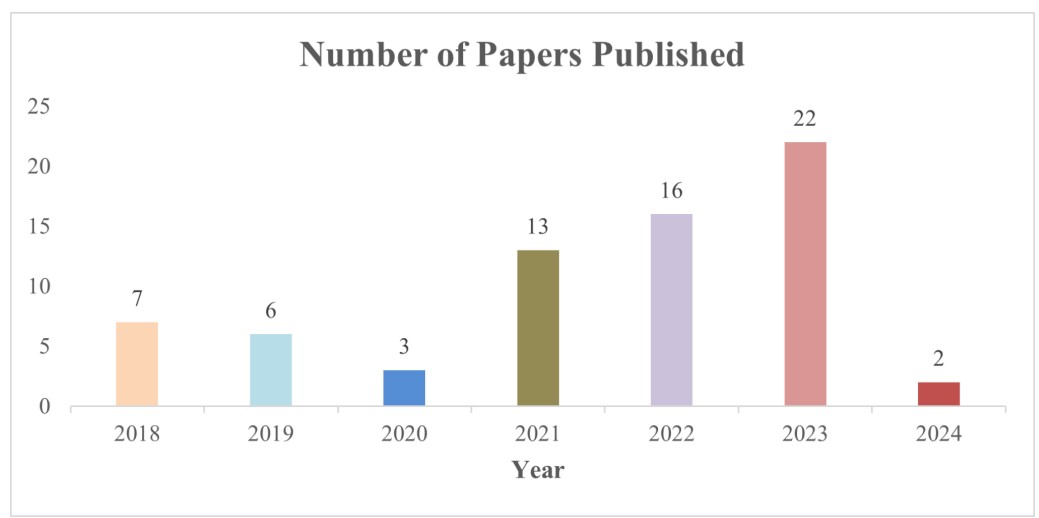

**Figure 2** **The number of papers published from 2018 to 2024.** The increasing research interest in multi-modal healthcare data fusion starting from 2021.

papers and authors. VOSviewer also helped visualize keyword co-occurrences, revealing research trends and clusters.

Visual representation in Fig. 3 highlights the main words of the selected literature, using the abstract, title, and keywords of papers from the Web of Science (WOS) database. Analyzing with VOSviewer revealed 5 clusters (yellow, blue, green, red, and purple), showing relationships between topics. The yellow cluster focuses on deep learning techniques, including multimodal fusion, ensemble learning, multitask learning, and applications like sentiment analysis and remote sensing. The green cluster emphasizes multimodal learning and data fusion, encompassing machine learning techniques, neural networks, and classification algorithms. The purple cluster centers on feature extraction, visualization, and computational modeling, with an emphasis on attention mechanisms and task analysis. The red cluster highlights artificial intelligence applications in cancer research, including predictive models and deep learning approaches. The blue cluster underscores the integration of multimodal data and ensemble learning techniques, focusing on prediction and data fusion strategies. By analyzing the keywords within each cluster, we gain insights into key themes, trends, and research directions, informing further investigation and collaboration.

## RESULTS

### Overview of different fusion techniques for multimodal data for medical applications

Data fusion integrates various data types to address inference problems by combining different viewpoints on a phenomenon. This technique leverages features within different data sources to refine estimates and predictions (*Mohsen et al., 2022*). Combining data from multiple sources, often termed data fusion in biomedical literature, minimizes errors

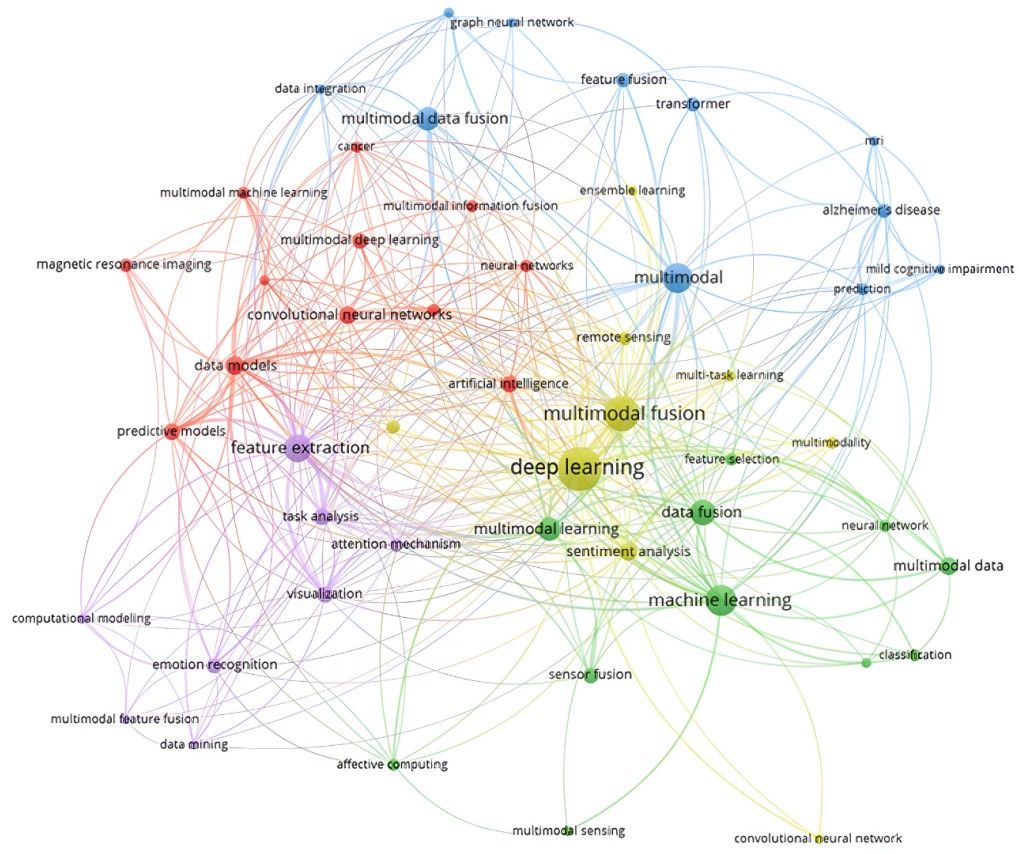

**Figure 3** Visual representation of the scientific landscape of the selected studies using VOSviewer's mapping function.

compared to single-source approaches (*Stahlschmidt, Ulfenborg & Synnergren, 2022*). The primary goal is to extract and integrate complementary contextual information from diverse sources to facilitate decision-making. This approach allows AI models to use information from various sources, particularly beneficial with noisy or incomplete data, enhancing robustness and accuracy (*Lipkova et al., 2022*). There are three main types of data fusion techniques: early, intermediate or joint, and late fusion, as illustrated in Fig. 4. Early fusion, also known as feature-level or low-level fusion, consolidates multiple input modalities into a unified feature vector before training a single machine learning model. This process uses methods like concatenation, pooling, or gated units to merge input modalities. There are two primary types: type I combines original features, while type II integrates extracted features from methods such as manual techniques, imaging software, or other neural networks (*Huang et al., 2020a*; *Moshawrab et al., 2023*). Early fusion merges modalities based on predictor information or independent variables, serving either as a preprocessing step or an unsupervised task to create features that capture underlying patterns (*Gaw, Yousefi & Gahrooei, 2022*; *Stahlschmidt, Ulfenborg & Synnergren, 2022*). While early fusion strategies are effective at learning relationships across modalities from

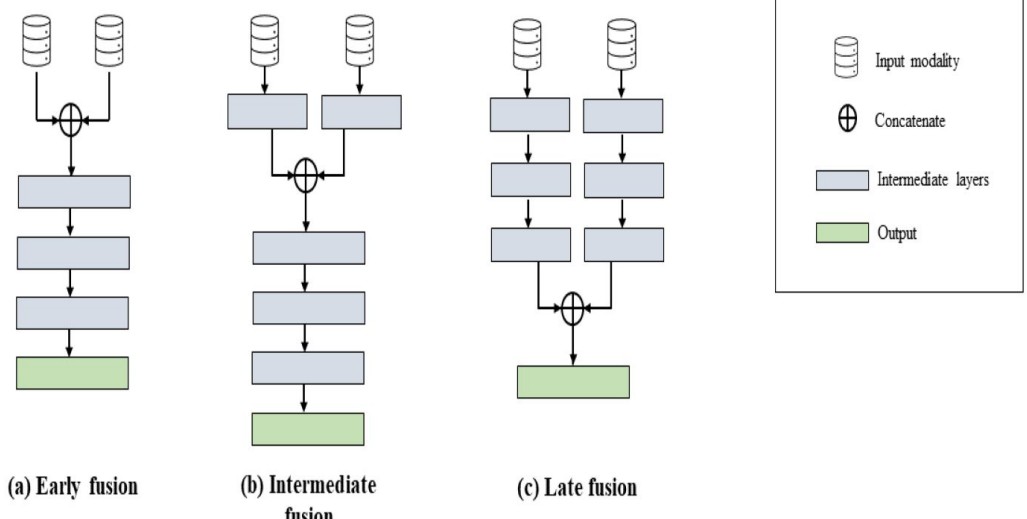

**Figure 4** Illustrations of architectures of different fusion techniques.

low-level features, they may not capture higher-level relationships that require explicit learning of marginal representations. These strategies can also be sensitive to variations in sampling rates among modalities (*Stahlschmidt, Ulfenborg & Synnergren, 2022*).

Intermediate fusion, also known as joint or middle fusion, integrates learned feature representations from intermediate layers of neural networks with features from different modalities. Unlike early fusion, intermediate fusion allows the loss during training to influence feature extraction models, refining representations iteratively (*Huang et al., 2020a*). This approach focuses on learned feature representations rather than original multimodal data, enabling neural networks to learn these representations, whether homogeneously or heterogeneously designed. This can potentially discover more informative latent factors (*Stahlschmidt, Ulfenborg & Synnergren, 2022*). It is often demonstrated through branched neural network models that merge learned feature representations from intermediate layers with other source features, enhancing the model's understanding of combined representations (*El-Ateif & Idri, 2022*; *Shetty, Ananthanarayana & Mahale, 2023*).

Late fusion, or decision-level fusion, consolidates predictions from multiple models into a final decision. This process involves training separate models for different modalities and then employing an aggregation function to merge these models' predictions (*Huang et al., 2020a*). It utilizes different rules, like Max-fusion, Averaged-fusion, or Bayesian rules, to fuse decisions from distinct classifiers (*Moshawrab et al., 2023*). Late fusion integrates feature vectors from individual modalities *via* separate discriminative models, combining resulting probability values into final feature vectors for each patient. This process incorporates a meta-learner to weigh the significance of each prediction source rather than individual features, thereby enhancing the final label's accuracy (*El-Sappagh et al., 2020*).

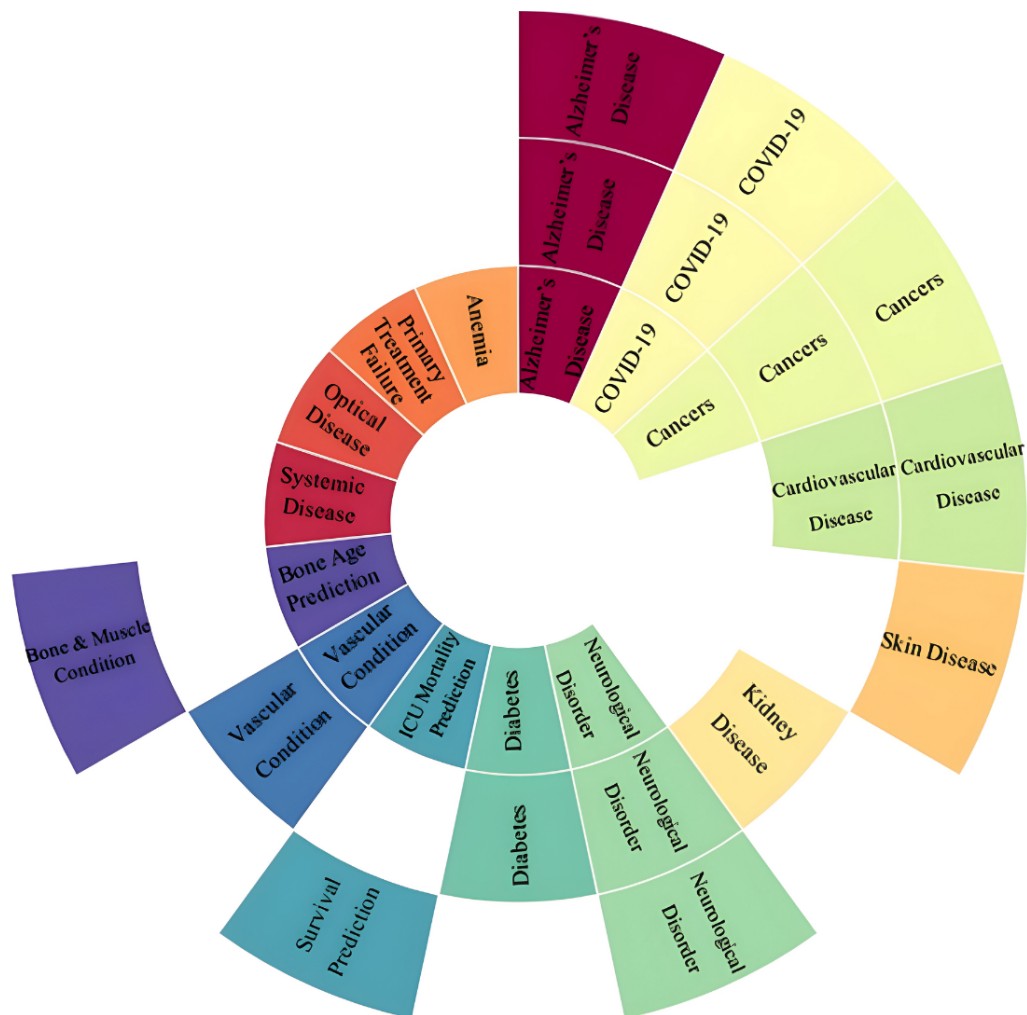

**Figure 5** **Sunburst chart that represents the types of fusion techniques used in different diseases.** Early fusion (inner ring), intermediate or joint fusion (middle ring), and late fusion (outer ring).

## Different fusion techniques in healthcare AI model development

The application of early, intermediate, and late fusion in medical condition in various diseases such as Alzheimer's disease (AD), anemia, various cancer type and many more as shown in Fig. 5. Research on data fusion techniques in the medical field and for various diseases has progressed significantly from 2018 to the present. The highlights and gaps are summarized in Table 1. It is evident from Table 1 that Alzheimer's disease and various types of cancer have the highest number of published papers. Most studies emphasize the advantages of using multimodal data to develop advanced models for disease detection and prediction. However, some limitations, such as missing data and small sample sizes, are also noted. These limitations and proposed future work are discussed in the Future Trends section.

**Table 1  Summary table for different types of fusion techniques based on different types of diseases.**

| Disease | Reference (Author) | Early | Joint | Late | Concluding remarks | Gap |
|---|---|---|---|---|---|---|
| Alzheimer | Bhagwat et al. (2018) | / | | | Most of the technical papers focused on developing prediction and classification models using multimodal data for Alzheimer's disease. One technical paper focused on developing a detection model. This model aimed to detect the presence of Alzheimer's disease and mild cognitive impairment. In summary, the technical papers examined in this review collectively highlight the advancements in machine learning and deep learning approaches for Alzheimer's disease research using data fusion. Some studies achieved significant improvement with approximately 90% accuracy, precision, AUC, recall, and F1-score after employed multimodal data. The consistent use of the ADNI dataset across multiple studies underscores its significance as a valuable resource for Alzheimer's disease research. There are few studies implemented other datasets for increased and validation of model such as AIBL, PPMI and colorectal cancer dataset. | Challenges include feature selection consistency and missing timepoints. |
| | Dimitriadis et al. (2018) | / | | | | Limited cohort size for trajectory modeling. |
| | Li & Fan (2019) | / | | | | |
| | Dai et al. (2021) | / | | | | |
| | Chen et al. (2023) | / | | | | |
| | Li et al. (2023a) | / | | | | |
| | Odusami et al. (2023) | / | | | | |
| | Spasov et al. (2018) | | / | | | Inability to consistently predict with 100% accuracy due to outliers. |
| | Lin et al. (2020) | | / | | | Missing data. |
| | Abdelaziz, Wang & Elazab (2021) | | / | | | Difficulty in finding ground truth labels for genetic data. |
| | Golovanevsky, Eickhoff & Singh (2022) | | / | | | |
| | Rahim et al. (2023b) | | / | | | |
| | Rahim et al. (2023a) | | / | | | |
| | Kadri et al. (2023) | | / | | | |
| | Lu et al. (2024) | | / | | | |
| | Feng et al. (2019) | | | / | | Limited availability of MRI and PET imaging data. |
| | Tang et al. (2023) | | | / | | |
| Anemia | Purwar et al. (2020) | / | | | The study proposed a method that combines blood smear image features extracted by a deep CNN and clinical features and achieved accuracy, sensitivity, and specificity of 99%, 1.00, and 0.98, respectively. | Limited sample size. |
| | | | | | | No comparison with existing diagnostic approaches. |
| Vascular condition | Liu et al. (2018) | / | | | The study employed a deep learning approach with multimodal data fusion to develop prediction model. The proposed methods make well prediction. One of the studies achieved overall prediction accuracy of 94.8% and another study predicted severe hemorrhages better than human experts and machine learning models that utilized single data modality. | – |
| | Akazawa & Hashimoto (2023) | | / | | | MRI image segmentation is not efficient. |
| | | | | | | Small sample size. |

Teoh et al. (2024), *PeerJ Comput. Sci.*, DOI 10.7717/peerj-cs.2298

**Table 1** (*continued*)

| Disease | Reference (Author) | Early | Joint | Late | Concluding remarks | Gap |
|---|---|---|---|---|---|---|
| Covid 19 | Kumar et al. (2022a) | / | | | Most of the studies employed deep learning in multimodal data fusion. Remarkably, most of these studies achieved exceptional accuracy rates of 90% or higher following the integration of multimodal data. This outcome highlights the potential of multimodal data fusion techniques to enhance predictive accuracy and diagnostic capabilities in Covid-19 research. The studies that employed early fusion utilized public datasets in their research and others utilized private datasets. | – |
| | Kumar et al. (2022b) | / | | | | |
| | Zhou et al. (2022) | | / | | | Lacks real-world application validation. |
| | Dipaola et al. (2023) | | / | | | Algorithm efficiency and manual revision requirements for optimization were limitations. |
| | Xu et al. (2021) | | | / | | Lack of exploration on the effectiveness of clinical data for diagnosis. |
| | Zheng et al. (2021) | | | / | | Integration challenges between high-dimensional CT imaging and low-dimensional features. |
| | Zhang et al. (2022) | | | / | | Missing data/ data inadequacy |
| Cancer | Kharazmi et al. (2018) | / | | | These studies focused on various cancer types including breast, skin, lung, and prostate, and it is evident that deep-learning methodologies have been extensively utilized for prediction, classification, and detection tasks. These studies focused on various cancer types including breast, skin, lung, and prostate, and deep-learning methodologies have been extensively used for prediction, classification, and detection tasks. Many studies reported accuracy, sensitivity, specificity, and AUC values of 0.80 and above when employing data fusion techniques. This showed the effectiveness of multimodal data fusion in enhancing predictive and diagnostic capabilities. | Limited demographic and tumor-related features used in the model. |
| | Nie et al. (2019) | / | | | | Small datasets |
| | Silva & Rohr (2020) | / | | | | Limited clinical information and small number of patients in previous studies. |
| | Mokni et al. (2021) | / | | | | Sample size imbalance. |
| | Yan et al. (2021) | / | | | | Potential biases from missing data. |
| | Joo et al. (2021) | / | | | | |
| | Tan et al. (2022) | / | | | | |
| | Oh et al. (2023) | / | | | | |
| | Yala et al. (2019) | | / | | | Limited sample size. |
| | Wang et al. (2021) | | / | | | Do not have independent datasets for validation. |
| | Schulz et al. (2021) | | / | | | Lack of generalization ability of external datasets for model validation. |
| | Qiu et al. (2022) | | / | | | |
| | Yao et al. (2022) | | / | | | |
| | Wei et al. (2023) | | / | | | |

**Table 1** (*continued*)

| Disease | Reference (Author) | Early | Joint | Late | Concluding remarks | Gap |
|---|---|---|---|---|---|---|
| | *Reda et al. (2018)* | | | / | | Lack of validation datasets. |
| | *Fu et al. (2021)* | | | / | | Small sample size. |
| | *Yang et al. (2022)* | | | / | | |
| | *Caruso et al. (2022)* | | | / | | |
| | *Holste et al. (2023)* | | | / | | |
| ICU mortality prediction | *Lin et al. (2021)* | / | | | The study proposed deep learning with multimodal data for ICU mortality prediction and the results demonstrated notable improvements in C-index, with values of 0.7847. | Missing value in datasets. |
| Diabetes | *Hsu et al. (2021)* | / | | | The studies focused on multimodal data fusion for diabetes prediction and classification using deep learning techniques. By integrating multiple data modalities, including EHR and imaging data, they achieved notable improvements in accuracy compared to single-modality approaches. | Limited by the availability of EHR data. |
| | *Hu et al. (2023)* | / | | | | |
| | *El-Ateif & Idri (2022)* | | / | | | – |
| Diffuse large B-cell lymphoma (DLBCL) | *Yuan et al. (2023)* | / | | | This study constructed multimodal deep learning by integrating multiple image modalities and EHR. It achieved 91.22% and 0.925 of accuracy and AUC after optimization of model. | – |
| Optical disease | *Chaganti et al. (2019)* | / | | | Both studies employed CNN for prediction and detection task using early fusion. The merged CNN for the first study achieved AUC of 0.74 while another study achieved AUC of 0.9796. However, both studies showed improvement after implementing multimodal data. | The study lacks direct segmentation for optic nerve volume estimation. |
| | *Jin et al. (2022)* | / | | | | |
| | *Zhang et al. (2023)* | / | | | | Data insufficiency or model simplicity. |
| Neurological disorder | *Yoo et al. (2019)* | | / | | These studies worked on multimodal deep learning by integrating MRI and clinical data. The results showed significant improvements in AUC and accuracy. | Limited training samples. |
| | *Huang et al. (2022)* | | | / | | – |
| Bone and muscle | *Li et al. (2023b)* | / | | | The authors reported that multimodal deep learning models outperformed the traditional approach with improved accuracy, sensitivity, and AUC. | – |
| | *Jujjavarapu et al. (2023)* | | | / | | Deep learning is computationally expensive and less interpretable. |
| | *Schilcher et al. (2024)* | | | / | | |
| Systemic disease | *Zhao et al. (2022)* | / | | | The author developed a multimodal deep learning method to predict systemic diseases using oral condition. The best accuracy and AUC achieved by the model are 0.92 and 0.88 respectively. | Limited generalizability. |
| Cardiovascular disease | *Yao et al. (2023)* | | / | | The studies highlighted that the use of multimodal deep learning architectures demonstrates superior performance compared to unimodal approaches, showcasing the importance of integrating multiple healthcare data such as chest X-ray images, fundus images, ECG data and EHR. The significant improvements shown in the high accuracy and AUC of models were achieved. | Small dataset. |
| | *Lee et al. (2023)* | | / | | | |
| | *Hsieh et al. (2023)* | | / | | | |
| | *Puyol-Anton et al. (2022)* | | | / | | Lack of testing with the same pipeline for predicting response to other types of treatment. |
| | *Jacenków, O'Neil & Tsaftaris (2022)* | | | / | | |

**Table 1** (*continued*)

| Disease | Reference (Author) | Early | Joint | Late | Concluding remarks | Gap |
|---|---|---|---|---|---|---|
| Kidney disease | *Chen et al. (2023)* | | / | | The author reported that fusion technique improved sensitivity (0.822) in detecting hyperplastic parathyroid glands for chronic kidney disease. | Lack of spatial information. |
| | | | | | | False-positive results. |
| Survival prediction | *Chen et al. (2021)* | | | / | The proposed method consistently outperformed state-of-the-art methods in survival outcome prediction in computational pathology, achieving superior performance with a 3.0% to 6.87% in overall C-Index. | Using previously curated gene set with potentially overlapping biological functional impact. |
| Skin disease | *Cai et al. (2023)* | / | | | Both studies proposed multimodal deep learning classification model that outperformed a baseline method. They combined multiple imaging modalities such as dermatoscopic images and macroscopic images with patient metadata for skin lesion classification. | – |
| | *Yap, Yolland & Tschandl (2018)* | | | / | | No comparison with human physicians limits clinical relevance insights. |

## Early fusion

A total of 69 studies were identified in this research endeavor. The predominant utilization of early fusion (28/69) and intermediate fusion (23/69) methodologies was observed in integrating multimodal healthcare data, with a comparatively lesser emphasis on late fusion (18/69). Among these twenty-eight early fusion studies, seven focused on diagnosing and predicting Alzheimer's disease, while six were dedicated to cancer classification, with the remaining studies addressing other diseases.

For instance, among seven studies in AD, there are two significant studies which employed longitudinal data for disease predictions. *Bhagwat et al. (2018)* focused on the prediction of clinical symptoms trajectories of AD by training a Longitudinal Siamese Neural Network (LSN) on longitudinal multimodal data. The author successfully applied cross-validation using three different ADNI cohorts and achieved generalizability on validation dataset of AIBL dataset. The LSN achieved 0.900 accuracy and 0.968 AUC on ADNI datasets and achieved 0.724 accuracy and 0.883 AUC on the replication AIBL dataset. The study showed potential improvement in prognostic predictions and patient care in AD.

Various cancer types, such as breast cancer, lung cancer, skin cancer, and brain tumors, have employed multimodal data for AI development. For example, *Yan et al. (2021)* proposed deep learning architectures such as CNN and VGG-16 for breast cancer classification based on multimodal data. The author employed denoising autoencoder to increase low-dimensional structured EHRs data to high-dimensional so that it can be fed into the CNN with pathological images. The proposed method improved breast cancer classification accuracy up to 92.9%.

## Intermediate fusion

There are 23 studies related to multimodal fusion utilizing healthcare data. Among these, a predominant focus has been on Alzheimer's disease, exploring prediction models utilizing deep learning architectures. *Lin et al. (2020)* employed Extreme Learning Machine (ELM) with multiple modalities fusion to predict AD conversion within 3 years. In feature selection process, the author utilized the least absolute shrinkage and selection operator (LASSO) algorithm which had been said to be beneficial in selecting MRI features. Thus, the proposed model achieved 87.1% accuracy and AUC of 94.7 in predicting AD conversion.

In another study, *Akazawa & Hashimoto (2023)* employed pretrained VGG-16 for MRI image, alongside CNN for extracting features from both laboratory and demographic data. These features were then concatenated using a neural network to develop a prediction model for severe hemorrhage, surpassing the performance of human experts and single data type models.

Additionally, *Yoo et al. (2019)* implemented multimodal deep learning method in predicting multiple sclerosis conversion. The author combined user-defined MRI and clinical measurement in their proposed model and employed a technique called Euclidean distance transform to increase information density in multiple sclerosis lesion masks. The CNN-based prediction model achieved 75.0% accuracy in predicting disease activity

within two years and outperformed random forest model that only used user-defined measurements.

## Late fusion

In this section, a total of 18 studies have leveraged late fusion techniques in multimodal data fusion to develop detection and prediction models for various diseases. Notably, *Feng et al. (2019)* employed late fusion by incorporating primary features extracted from MRI and PET images into 3D-CNN deep learning architectures. Besides, the author applied FSBi-LSTM on hidden spatial information to enhance performance of model, resulting in enhanced diagnostic accuracy of 94.82%.

For prostate cancer diagnosis, *Reda et al. (2018)* employed late fusion techniques by concatenating outputs from diverse classifiers, integrating clinical biomarkers and extracted features from diffusion-weighted magnetic resonance imaging (DW-MRI). The early diagnosis system achieved 94.4% diagnosis accuracy with 88.9% sensitivity and 100% specificity on 18 DW-MRI datasets, indicating promising results for the computer-aided diagnostic system.

In the cardiovascular field, *Puyol-Anton et al. (2022)* developed a multimodal deep learning framework (MMDL) by using 2D Deep Canonical Correlation Analysis (DCCA) algorithm for Cardiac resynchronization therapy (CRT) response prediction. By combining multimodal data, they achieved a CRT response prediction accuracy of 77.38%, demonstrating that the MMDL classifier improves accuracy compared to baseline approaches. Overall, late fusion techniques have shown efficacy in enhancing disease detection and prediction models across diverse medical domains.

## Type of multimodal data

Health data categorizes into three main types: imaging, clinical, and omics data. Each category provides unique insights, but their fusion yields a fuller disease comprehension, reducing ambiguity and enhancing model efficiency in medical data analysis. Imaging methods like MRI, CT, PET, and SPECT offer varied perspectives on anatomy and physiology. Clinical data, including patient histories, age, gender, and medication records, aid clinicians in understanding patient characteristics and disease progression, enhancing the contextual understanding of patient health. Additionally, genetic data play a crucial role in predicting and diagnosing conditions, providing valuable insights into disease progression and individualized treatment (*Behrad & Abadeh, 2022*). The example types of unstructured and structured data are shown in Fig. 6.

## Fusion of image and structured data

The integration of medical images and EHR has emerged as a pivotal strategy in enhancing predictive modeling across diverse medical domains. An example of architectures in fusing medical images and structured data from EHR is shown in Fig. 7.

In recent years, there has been a growing body of research focused on utilizing medical images and EHRs in Alzheimer's disease research. In eight studies, *Bhagwat et al. (2018)*, *Spasov et al. (2018)*, *Dimitriadis et al. (2018)*, *Li & Fan (2019)*, *Golovanevsky, Eickhoff & Singh (2022)*, *Rahim et al. (2023b)*; *Rahim et al. (2023a)* and *Lu et al. (2024)*, they employed

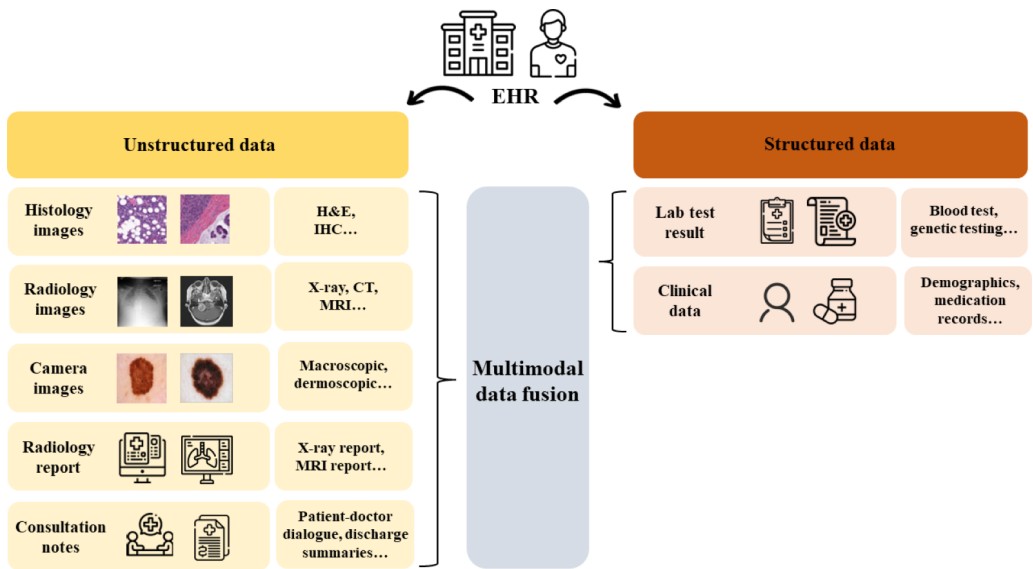

**Figure 6   Example types of unstructured and structured data in EHR.** Image sources: Hospital icon: https://www.freepik.com/icon/hospital_4320350; Patient icon: https://www.flaticon.com/free-icon/patient_1430402; Medical history icon: https://www.freepik.com/icon/medical-history_1424569; X-ray icon: https://www.freepik.com/icon/x-ray_7399390; Consultation icon: https://www.flaticon.com/free-icon/consultation_10202726?; Medical report icon: https://www.flaticon.com/free-icon/medical-report_3215528?; Medical checkup icon: https://www.flaticon.com/free-icon/medical-checkup_3061457?; Medicine icon: https://www.freepik.com/icon/medicine_4063711; People icon: https://www.veryicon.com/icons/miscellaneous/8atour/people-23.html; Medicine icon: https://www.freepik.com/icon/medicine_4063711; Lab result icon: https://www.flaticon.com/free-icon/invoice_751904; Histological images: https://www.kaggle.com/datasets/paultimothymooney/breast-histopathology-images/data; camera images (skin lesion): Tschandl P, Rosendahl C & Kittler H. The HAM10000 dataset, a large collection of multi-source dermatoscopic images of common pigmented skin lesions. Sci. Data 5: 180161 DOI: 10.1038/sdata.2018.161 (2018); X ray image: https://www.kaggle.com/datasets/financekim/curated-cxr-report-generation-dataset, Public Domain; brain MRI: https://www.kaggle.com/datasets/masoudnickparvar/brain-tumor-mri-dataset.

MRI in conjunction with clinical assessments, demographic details, and genetic data, particularly the APOe4 marker, from the Alzheimer's Disease Neuroimaging Initiative (ADNI) dataset. These technical papers have significantly contributed to the understanding and diagnosis of AD, as well as to the development of potential treatment strategies.

In the context of anemia, *Purwar et al. (2020)* undertook an innovative approach in anemia detection and prediction, integrating blood smear images with clinical data from complete blood count test (CBC) from AIIMS datasets. The features are extracted by deep CNN and fusion technique is applied. The dimensions of fused datasets are reduced by using linear discriminant analysis (LDA) and principal component analysis (PCA). The resulting model exhibited remarkable accuracy, reaching a maximum of 99%.

In addition, *Puyol-Anton et al. (2022)* explored cardiovascular magnetic resonance images (CMR) and electrocardiogram (ECG) data from UK Biobank and EchoNet-Dynamic to predict the response to cardiac resynchronization therapy (CRT) for heart failure patients. The combination of medical images and healthcare data enables CRT

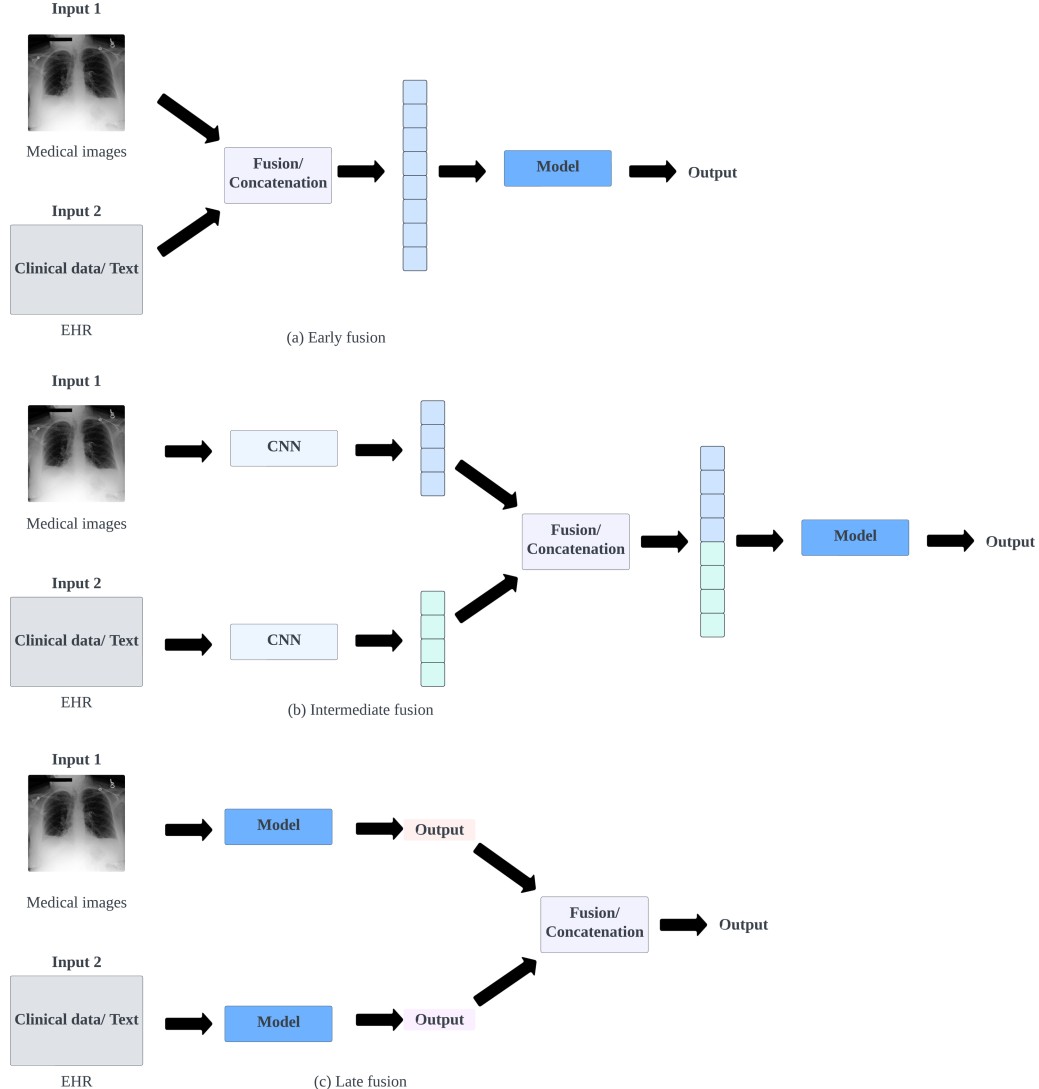

**Figure 7  Type of data fusion framework for image and structure data using early, intermediate, and late fusion.** X ray images: https://www.kaggle.com/datasets/financekim/curated-cxr-report-generation-dataset, Public Domain.

response prediction with 77.38% accuracy which is comparable with the current state-of-the-art in machine learning-based CRT response prediction.

## Fusion of image and unstructured data

In Covid-19, *Kumar et al. (2022a)*; *Kumar et al. (2022b)* conducted pioneering research by exploring healthcare data fusion techniques involving chest X-ray images and audio data of cough for early diagnosis and accurate classification of Covid-19 cases. They utilized several public datasets such as IEEE-8023 CXR—Cohen dataset, Coswara, Coughvid and many more as stated in Table 2. Additionally, *Zheng et al. (2021)* employed a comprehensive

approach by fusing various image modalities, including X-ray, CT scan, and ultrasound images, with medical consultation data to enhance the classification of COVID-19 cases.

*Jacenków, O'Neil & Tsaftaris (2022)* demonstrated an innovative application of data fusion by combining chest X-ray images with their corresponding radiological reports to develop a classification model for cardiovascular diseases. This approach integrates radiological images with textual information from MIMIC CXR, enriching the diagnostic process for cardiovascular conditions. By integrating different modalities, the proposed method achieved an average micro AUROC of 87.8, outperforming the state-of-the-art methods for unimodal of 84.4 AUROC.

## Fusion of multiple types of images

Several studies have explored image fusion techniques for Alzheimer's disease detection. Noteworthy contributions include the work of *Dai et al. (2021)*, which advocate for merging Positron Emission Tomography (PET) with Magnetic Resonance Imaging (MRI) from ADNI and PPMI datasets. The paper proposed a classification model for Alzheimer's disease diagnosis based on improved CNN models and image fusion method, achieving high AUC values of 0.941 in training with fusion images. The research demonstrated that the proposed method using fusion images dataset based on multi-modality images has higher diagnosis accuracy than single modality images dataset. Meanwhile, *Kadri et al. (2023)* further extended the exploration by combining MRI with both PET and CT, showcasing the versatility of data fusion in Alzheimer's disease diagnosis.

Within the domain of cancer research, *Mokni et al. (2021)* conducted a notable study wherein they employed data fusion techniques. Specifically, they integrated Dynamic Contrast-Enhanced Magnetic Resonance Imaging (DCE-MRI) with Digital Mammographic images (MGs) for the purpose of detecting breast cancer. The author extracted the features by using Gradient Local Information Pattern (GLIP) and performed Canonical Correlation Analysis (CCA) for multimodal fusion. The proposed method achieved an AUC value of 99.10% compared to AUC values for MG and DCE-MRI modalities alone of 97.20% and 93.50%, respectively. This integrative approach capitalizes on the complementary strengths of DCE-MRI and MGs, offering a more comprehensive and detailed insight into breast cancer characteristics, ultimately contributing to improved diagnostic accuracy.

## Other fusion features

In a significant contribution to the field of vascular conditions, *Liu et al. (2018)* conducted a comprehensive study focusing on the prediction of anterior communicating artery (ACOM) aneurysms. The research involved the integration of diverse healthcare data, including CT images, EHR, and textual reports. By combining these various sources of information, the study aimed to enhance the accuracy and depth of predicting ACOM aneurysms, illustrating the potential of data fusion in advancing vascular condition diagnostics.

*Lin et al. (2021)* made notable strides in predicting mortality rates in Intensive Care Units (ICUs) by exploring different healthcare data sources. The study integrated chest X-ray images, clinical data, and radiological reports from MIMIC IV to develop a robust

**Table 2** Summary table of public datasets used by each study.

| Disease | Type of data | Dataset used | Reference |
|---------|-------------|-------------|-----------|
| Alzheimer disease | PET, CT, MRI, Age, gender, education years, APOE $\varepsilon 4$ status at baseline, cerebrospinal fluid biomarkers, gene data, cognitive scores, | ADNI https://adni.loni.usc.edu/data-samples/access-data/ | *Bhagwat et al. (2018)* |
| | | | *Dimitriadis et al. (2018)* |
| | | | *Li & Fan (2019)* |
| | | | *Dai et al. (2021)* |
| | | | *Chen et al. (2023)* |
| | | | *Li et al. (2023a)* |
| | | | *Odusami et al. (2023)* |
| | | | *Spasov et al. (2018)* |
| | | | *Lin et al. (2020)* |
| | | | *Abdelaziz, Wang & Elazab (2021)* |
| | | | *Golovanevsky, Eickhoff & Singh (2022)* |
| | | | *Rahim et al. (2023b)* |
| | | | *Rahim et al. (2023a)* |
| | | | *Kadri et al. (2023)* |
| | | | *Lu et al. (2024)* |
| | | OASIS https://www.oasis-brains.org/ | *Kadri et al. (2023)* |
| | | PPMI https://www.ppmi-info.org/access-data-specimens/data | *Dai et al. (2021)* |
| | | AIBL http://adni.loni.usc.edu/category/aibl-study-data/ | *Bhagwat et al. (2018)* |
| Bone age assessment | X-rays (key bone regions), gender | RSNA dataset https://www.rsna.org/rsnai/ai-image-challenge/RSNA-Pediatric-Bone-Age-Challenge-2017 | *Li et al. (2023b)* |
| Microcytic hypochromia (Anemia) | Blood smear image, clinical features | AIIMS https://www.bioailab.org/datasets | *Purwar et al. (2020)* |
| Cancer prediction | Microscopy slides, Clinical data (cancer type, gender, race, history of prior malignancy, and age) | | *Silva & Rohr (2020)* |
| Survival outcome prediction | Whole slides images, genomic data | | *Chen et al. (2021)* |
| Prediction of HER2-positive breast cancer recurrence and metastasis risk | Whole slide H&E images (WSIs) and clinical information | TGCA https://portal.gdc.cancer.gov/ | *Yang et al. (2022)* |
| Colorectal cancer | Pathological images, multi-omic data | | *Qiu et al. (2022)* |

**Table 2** (*continued*)

| Disease | Type of data | Dataset used | Reference |
|---|---|---|---|
| Renal cancer | Histopathological images, CT/MRI scans, and genomic data from whole exome sequencing | KIRC TCGA (Kidney renal clear cell carcinoma of the Cancer Genome Atlas) GDC portal https://portal.gdc.cancer.gov/ cancer imaging archive https://www.cancerimagingarchive.net/ | *Schulz et al. (2021)* |
| Covid | Chest X-ray and cough sample data | IEEE-8023 CXR—Cohen dataset https://github.com/ieee8023/covid-chestxray-dataset Shenzhen CXR with Masks https://www.kaggle.com/datasets/yoctoman/shcxr-lung-mask Montgomery county CXR images https://paperswithcode.com/dataset/montgomery-county-x-ray-set COVIDGR 1.0 https://paperswithcode.com/dataset/covidgr | *Kumar et al. (2022a)* |
| | | Coswara https://github.com/iiscleap/Coswara-Data Coughvid https://cs.paperswithcode.com/paper/the-coughvid-crowdsourcing-dataset-a-corpus DetectNow https://github.com/shresthagrawal/detect-now Virufy https://github.com/virufy/virufy-data | *Kumar et al. (2022b)* |
| ICU-mortality prediction | Chest X-ray, clinical data (EHR), radiology reports | MIMIC IV https://physionet.org/content/mimiciv/2.2/ | *Lin et al. (2021)* |
| Diabetes | Fundus and WGBF | APTOS 2019 blindness detection https://www.kaggle.com/c/aptos2019-blindness-detection/data Messidor-2 https://www.kaggle.com/datasets/mariaherrerot/messidor2preprocess/data | *El-Ateif & Idri (2022)* |
| Breast cancer | Pathological images, EMR | PathoEMR dataset no longer available | *Yan et al. (2021)* |
| | Whole-slide images and gene expression profiles | TGCA https://portal.gdc.cancer.gov/ | *Wang et al. (2021)* |
| Cardiovascular disease | Chest X-rays, report | MIMIC CXR https://physionet.org/content/mimic-cxr/2.0.0/ | *Jacenków, O'Neil & Tsaftaris (2022)* |
| | Clinical risk factors and fundus photographs | UK Biobank https://www.ukbiobank.ac.uk/enable-your-research/apply-for-access | *Lee et al. (2023)* |
| Cardiac resynchronization therapy response prediction | CMR imaging, ECG data | UK Biobank (UKBB) https://www.ukbiobank.ac.uk/enable-your-research/apply-for-access EchoNet-Dynamic https://echonet.github.io/dynamic/ | *Puyol-Anton et al. (2022)* |
| Disease location in chest X-ray images | Chest X-ray, clinical data | MIMIC-CXR https://physionet.org/content/mimic-cxr/2.0.0/ MIMIC IV https://physionet.org/content/mimiciv/2.2/ REFLACX https://paperswithcode.com/dataset/reflacx | *Hsieh et al. (2023)* |
| Prostate cancer | MRI, clinical biomarkers | DW-MRI https://data.mendeley.com/datasets/fgf86jdfg6/1 | *Reda et al. (2018)* |
| Lung cancer | CT image, lung tumor biomarker | LCID https://wiki.cancerimagingarchive.net/pages/viewpage.action?pageId=1966254 | *Fu et al. (2021)* |
| | CT, clinical data | CLARO https://paperswithcode.com/dataset/claro | *Caruso et al. (2022)* |

model for predicting mortality in the ICU. The contributions of labels, text, and image features are demonstrated as shown in the C-index of the model achieved which is 0.7847, surpassing the baseline model.

_Zhang et al. (2023)_ made significant advancements in the detection of multiple sclerosis (MS), a neurological disorder by involving the fusion of various data sources, including brain MRI images, EHR, and free-text reports from patients' clinical notes. The proposed method successfully predicts MS severity with an increase of 19% AUROC. This comprehensive fusion of structured and unstructured data enables a more accurate prediction of multiple sclerosis, showcasing the potential of data integration in advancing neurological disorder prediction

## DISCUSSION

In the previous sections, a comprehensive review of recent studies from 2018 to the present focused on machine learning and deep learning techniques for diagnosing, prognosing, and predicting treatments for various diseases. The data fusion combinations are categorized into fusion of medical images with structured data, fusion of medical images with unstructured data, fusion of multiple image modalities, and other features fusion. Additionally, the data fusion techniques were classified into early, intermediate, and late fusion approaches.

Our analysis revealed that multimodal data fusion models consistently outperformed single-modality models across performance metrics such as accuracy, sensitivity, precision, AUC, and C-index. Therefore, it is recommended to employ a multimodal machine learning or deep learning model when multiple healthcare data sources are available, as incorporating additional clinical data from EHR often results in improved performance.

Figure 8 shows the proposed framework of this study for improving clinical decision support using multimodal data integration. The framework follows a cyclical pattern that begins with the collection of data from various hospitals or health centers. This data is then aggregated through multimodal data fusion and undergoes AI modeling processes. The algorithms analyze the data to extract valuable insights related to health outcomes, including diagnosis, prognosis, risk assessment, and treatment planning. These insights are communicated back to hospitals and practitioners, enabling informed decisions for patients.

The proposed framework for multimodal data integration in clinical decision support offers promising solutions to address several challenges faced by the current healthcare sector. One significant challenge lies in the limited availability of healthcare data such as medical images, clinical data and EHRs (_Feng et al., 2019_; _Hsu et al., 2021_; _Nie et al., 2019_; _Zhang et al., 2023_). By incorporating advanced data integration techniques, the proposed framework enables the integration of diverse types of healthcare data sources, thereby enhancing access to comprehensive and longitudinal patient health records. This facilitates more accurate diagnoses, enabling more informed clinical decision-making and personalized treatment strategies.

The lack of real-world application and the absence of comparison with human physicians in current healthcare practices represent another critical challenge that the proposed
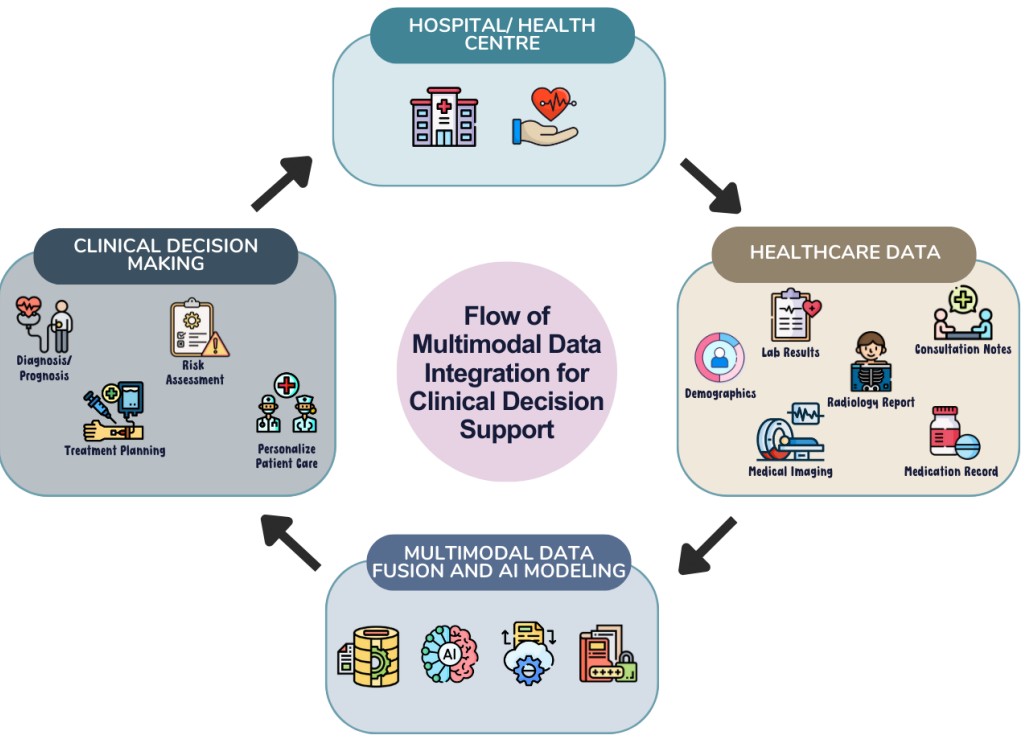

**Figure 8** **Proposed flow of multimodal data integration for clinical decision support.** Image sources: Hospital icon: https://www.freepik.com/icon/hospital_4320350; Cardiogram icon: https://www.flaticon.com/free-icon/cardiogram_7918446; Demographics icon: https://www.freepik.com/icon/demographics_2720724; CT scan icon: https://www.flaticon.com/free-icon/ct-scan_2355587; Aspirin icon: https://www.freepik.com/icon/aspirin_4320310; X-rays icon: https://www.flaticon.com/free-icon/x-rays_706196; Consultation icon: https://www.flaticon.com/free-icon/consultation_10202688; Diagnosis icon: https://www.freepik.com/icon/diagnosis_4320491; Risk management icon: https://www.flaticon.com/free-icon/risk-management_10240181; Diagnosis icon: https://www.flaticon.com/free-icon/diagnosis_1934430; Treatment icon: https://www.freepik.com/icon/treatment_3027624; Medical staff icon: https://www.flaticon.com/free-icon/medical-staff_6190390; Data processing icon: https://www.flaticon.com/free-icon/data-processing_2980479; AI icon: https://www.flaticon.com/free-icon/ai_8131880; Data synchronization icon: https://www.flaticon.com/free-icon/data-synchronization_4403186; Data encryption icon: https://www.flaticon.com/free-icon/data-encryption_4736094.

framework seeks to overcome (*Xu et al., 2021*; *Yap, Yolland & Tschandl, 2018*; *Zhou et al., 2022*). The proposed framework addresses this limitation by enabling the integration of expert knowledge and clinical guidelines into decision support systems, thereby facilitating comparative analyses between algorithmic predictions and human expert judgments. This not only enhances the interpretability and trustworthiness of algorithmic recommendations but also encourages collaboration between clinicians and data scientists in optimizing clinical decision-making processes.

## Future trends

Future work aimed at addressing the research gap identified in related studies should prioritize several key areas to enhance the field of healthcare data fusion and multimodal deep learning for clinical decision support. These include dealing with noisy or irrelevant

data that may impact model performance, as well as addressing issues related to missing or sparse data (*Bhagwat et al., 2018*; *Lin et al., 2021*; *Spasov et al., 2018*; *Tan et al., 2022*; *Zheng et al., 2021*). To tackle this, future research efforts should incorporate robust data imputation methods to address missing data issues effectively. Basic imputation techniques like k-nearest neighbors (KNN) can provide a foundation, while more advanced methods such as matrix completion and deep learning-based approaches can be explored to accurately estimate missing values and improve the quality of input data.

Besides, a significant barrier to the clinical implementation of multimodal deep learning methods is the limited availability of data (*Feng et al., 2019*; *Hsu et al., 2021*; *Joo et al., 2021*; *Nie et al., 2019*; *Zhang et al., 2023*). To address this issue, future researchers can implement data synthesis models that can learn the underlying data distribution and generate realistic data samples. The example of the models are generative adversarial networks (GANs) or variational autoencoders (VAEs). However, it is important to note that GANs and VAEs might produce augmented data that significantly differs from the raw data, potentially affecting model performance. When there is limited labeled data, semi-supervised learning is suggested. By exploring semi-supervised learning, models can be trained with both labeled and unlabeled data, effectively utilizing limited labeled data and unlabeled data to improve model performance.

Not only this, enhancing data sharing practices and improving access to comprehensive datasets are crucial steps toward advancing research in this field. Collaborative data-sharing platforms and standardized data collection protocols can help mitigate this challenge. By fostering a more open and cooperative data-sharing environment, researchers can gain access to the necessary resources to develop and validate more robust multimodal integration models. This, in turn, can lead to improved diagnostic accuracy and patient outcomes, ultimately benefiting the healthcare sector.

In addition, the presence of outliers in the data can significantly impact model performance (*Spasov et al., 2018*). Thus, future research should prioritize data preprocessing techniques aimed at detecting and removing outliers from the dataset before model training. Outlier detection techniques such as $z$-score, isolation forest, or k-nearest neighbors can be employed to identify and remove outliers from the dataset before training the model. After removing outliers from the data, another challenge arises which is integration difficulties between high-dimensional medical images and low-dimensional EHR features (*Xu et al., 2021*). To tackle this challenge, future work should explore dimensionality reduction techniques such as PCA and autoencoders. These techniques can be employed to achieve dimensionality reduction while preserving the discriminative power of the data.

Therefore, future research efforts should focus on developing and improving data fusion methodologies that address challenges related to noisy or limited data, outlier detection, and integration difficulties. By overcoming these challenges, future frameworks for multimodal deep learning in clinical decision support can significantly enhance diagnostic accuracy, treatment efficacy, and patient outcomes in healthcare settings.

## CONCLUSION

This paper presents a comprehensive analysis of methods for integrating multiple types of data in artificial intelligence models for healthcare. Our review includes 69 relevant publications from 2018 to 2024, offering an in-depth investigation of fusion techniques, such as incorporating medical images with organized and unorganized data, merging distinct image modalities, and amalgamating diverse characteristics. We highlight the utilization of data fusion approaches for different diseases, demonstrating how customized fusion strategies can effectively address specific diagnostic and therapeutic challenges. Focusing on these diseases provides a clearer understanding of the practical benefits of combining multiple data types in therapeutic settings. Our extensive review of contemporary data fusion technologies and their applications is a valuable resource for scholars and practitioners. By outlining the advantages and constraints of each method, we provide direction for future research aimed at creating and enhancing multimodal AI models in healthcare. Data fusion technologies are continually advancing and hold great potential for the future of healthcare. Advancements in this domain could improve the resilience, effectiveness, and precision of AI systems, enhancing patient outcomes and propelling medical science forward. The integration of diverse healthcare data from multiple sources is crucial for the advancement of AI model development. This paper enhances current knowledge by combining previous literature and examining different fusion strategies, providing a comprehensive understanding of the subject and establishing a foundation for future research focused on utilizing multimodal data to develop better healthcare solutions.

### Funding

This project was supported by the Xuzhou Science and Technology Project under Grant No. KC21182 and the Universiti Malaya Matching Grant under the project code of MG004-2024. The funders had no role in study design, data collection and analysis, decision to publish, or preparation of the manuscript.

### Grant Disclosures

The following grant information was disclosed by the authors:
Xuzhou Science and Technology Project: Grant No. KC21182.
Universiti Malaya Matching Grant under the project code: MG004-2024.

### Competing Interests

The authors declare there are no competing interests.

### Author Contributions

- Jing Ru Teoh conceived and designed the experiments, performed the experiments, analyzed the data, performed the computation work, prepared figures and/or tables, authored or reviewed drafts of the article, and approved the final draft.

- Jian Dong conceived and designed the experiments, analyzed the data, performed the computation work, authored or reviewed drafts of the article, and approved the final draft.
- Xiaowei Zuo performed the experiments, analyzed the data, authored or reviewed drafts of the article, and approved the final draft.
- Khin Wee Lai conceived and designed the experiments, analyzed the data, authored or reviewed drafts of the article, and approved the final draft.
- Khairunnisa Hasikin performed the experiments, analyzed the data, performed the computation work, prepared figures and/or tables, authored or reviewed drafts of the article, and approved the final draft.
- Xiang Wu analyzed the data, prepared figures and/or tables, and approved the final draft.

### Data Availability

This is a literature review.

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
