# Peer review of "Advancing healthcare through multimodal data fusion: a comprehensive review of techniques and applications"

_PeerJ Computer Science, doi:10.7717/peerj-cs.2298_

## Round 0.1 · original submission · Major Revisions

Thank you for your submission to PeerJ Computer Science again. Jing Ru Teoh et. al. presented a review of using the multimodal data fusion with different types of healthcare data, inlcuing such EHR and image data. Besides the reviewers’ comment, I have listed my suggestions as below. Please feel free to let me know if you have any questions or concerns during the revision.

Major:
1. In the abstract, the authors might add a short summary of their main findings or the key highlights from their literature review. This summary should convey the most important takeaways and the overall significance of the study, allowing readers to quickly grasp the essence of the research.
2. It is recommended that the authors include a PRISMA Flow Diagram in their manuscript although there is no meta-analysis has been done. This diagram will provide a clear and visual representation of the article selection process.
Minor:
1. For Table 2, the authors may consider adding another column to indicate whether access to the databases mentioned is public or private.
2. Authors mentioned the limited availability of the healthcare data. This might need to be extended a little bit more towards to the data sharing and obtaining for the researchers.

Reviewer 1 ·

Basic reporting

The review paper on "Advancing Healthcare through Multimodal Data Fusion" addresses a broad and cross-disciplinary topic relevant to both the fields of healthcare and artificial intelligence. It significantly contributes to the growing interest in multimodal data fusion techniques in healthcare, showcasing its potential to enhance patient care, decision-making, and healthcare outcomes. The topic aligns well with the journal's scope, particularly concerning innovative applications of computer science in healthcare.

The field of multimodal data fusion in healthcare has been active, especially from 2018 onwards, as demonstrated by the increasing number of publications. However, this review is justified despite the recent reviews in the field, as it provides a fresh perspective by focusing on the integration of diverse data types (such as images, structured, and unstructured data) and detailing the application-specific benefits, which adds a new dimension to the existing literature.

Overall, the paper is well-crafted, providing a clear context and importance of multimodal data fusion in healthcare. It effectively outlines the limitations of current unimodal approaches and positions the review as a necessary synthesis to push the field forward. The target audience, primarily researchers and practitioners in biomedical engineering and healthcare informatics, is well defined, and the motivation for the review is compellingly articulated.

Experimental design

Survey Methodology: The survey methodology employed in the manuscript is designed to ensure comprehensive and unbiased coverage of the subject. It involves a systematic selection of relevant studies, focusing on those published from 2018 onwards, using the Web of Science database. This method is appropriate given the rapid advancements in the field of multimodal data fusion in healthcare.

However, there are a few areas that could potentially enhance the comprehensiveness of the review: 1) Database Selection: The exclusive use of Web of Science might limit the scope of the surveyed literature. Including additional databases like PubMed, IEEE Xplore, or Scopus could provide a broader spectrum of relevant research, especially from different interdisciplinary fields that contribute to healthcare and data fusion. 2) Search String Specificity: While the manuscript details the general search strings used, it could benefit from including more specific keywords or varied terminology used in multimodal data fusion to ensure no significant studies are overlooked.

Citation and Referencing: The manuscript appears to cite sources adequately, with a good mix of foundational and recent studies that underline the narrative of recent advancements in the field. Citations are used effectively to support claims, and both direct quotations and paraphrasing are appropriately used and referenced. This robust referencing underlines the manuscript's alignment with academic standards and adds credibility to the review.

Organizational Structure: The review is well-organized, and structured into logical sections that guide the reader through the complexities of multimodal data fusion in healthcare. The manuscript progresses coherently from the introduction of the topic, detailing the methodologies, discussing the applications, and finally, exploring future directions and conclusions. Each section is clearly defined with subheadings, and the flow of content within sections is smooth, facilitating an easy understanding of a complex subject. It would be great to visually present the logic flow at the beginning of the manuscript, not later.

Validity of the findings

Development and Support of Arguments: The manuscript effectively develops and supports its arguments, aligning with the goals outlined in the introduction. The central thesis is that the integration of multimodal data through advanced fusion techniques can significantly enhance diagnostic and prognostic capabilities in healthcare.

Identification of Unresolved Questions and Future Directions: The conclusion of the manuscript effectively identifies several unresolved questions and potential future directions, addressing the complexity and evolving nature of the field.

Overall, findings are valid.

Additional comments

Nice work. You can further improve the writing of the paper by incorporating English copy-editing -- it's better to keep the active/passive voice and tense consistent across the whole paper.

Reviewer 2 ·

Basic reporting

no comment

Experimental design

no comment

Validity of the findings

no comment

Additional comments

Reviewer comments for Manuscript peerj-reviewing-100144-v0

Dear editors,

Jing et al. conducted a comprehensive literature review on multimodal data fusion approaches for biomedical applications. They thoroughly discussed various fusion methods, including early, intermediate, and late fusion. The authors also statistically analyzed the data fusion applications from selected literature. While their efforts in reviewing and analyzing the literature are commendable, the paper lacks logical consistency and depth in its drafting and perspectives. Notably, there is a significant gap in the technical discussion of algorithms and deep learning models. Here are my detailed comments:

1. Language and Grammar Errors: (All the line indexes are based on the review file where Abstract caption is the 41st lines) There are multiple language and grammar issues that need correction. For example, the review contains repeated sentences (lines 66-69) and abbreviations (e.g., multiple occurrences of the fully name and abbreviation of HER), as well as punctuation errors (line 162 figure 1 uses a colon “:”, whereas line 201 figure 2 uses a semicolon “;”). These are just a few examples. The language should be thoroughly checked and corrected.

2. Logical Consistency: The paper’s logic is inconsistent. For instance, on line 73, the authors discuss the limitations of single modality and the importance of multimodality work. Then, on line 90, they restate the limitations of single modality methods. I suggest merging paragraphs with similar content to improve coherence.

3. Analysis Methodology: The authors analyze the field of fusion data in biomedical applications by prescreening the literature using specific keywords and metrics. However, different keywords and metrics could yield different statistical results. Additionally, merely analyzing the quantity of papers lacks meaning. A good review should technically summarize the development and gaps in the field by closely examining the content of all relevant papers, rather than relying solely on keyword-based screening and statistical analysis.

4. Terminology Consistency: Terminology should be consistent throughout the paper. For example, on line 281, the term “middle fusion” is used. This should be consistently referred to as “intermediate fusion” or “middle-level fusion”.

5. Definition Clarity: On line 404, the discussion on types of data fusion begins. However, previous sections have mixed the discussion of these data types. I suggest defining multimodal data and data fusion at the beginning of the paper for clarity.

6. Technical Details and Modeling: The technical details and modeling sections need more specification. For instance, on line 443, the authors mention a published work on the fusion of MRI images and tabular data, stating that a “multimodal deep learning model” was employed. As a review paper, it should detail the specific models and algorithms used, such as LAVA, GPT-4, or others.

7. Definition of Unstructured Data: On line 470, the authors discuss the fusion of images and unstructured data. The term “unstructured data” is broad and refers to various data types. It would be helpful to define unstructured data first.

8. Terminology Precision: On line 485, the term “multiple image modalities” is used. It might be more precise to say "multiple types of images" since "modalities" can refer to text, image, voice, etc. The term should be used appropriately.

9. Objective Tone: On line 558, the authors refer to “the framework of my study”. It would be better to adopt a more objective tone.

10. Future Work and Perspectives: In the future work and perspectives sections, like line 610, the authors mention “advanced techniques like KNN”. Considering KNN is not considered advanced in recent years, this could be modified. Additionally, while the authors suggest GANs and VAEs for data augmentation to address data limitations, it should also be noted that these approaches sometimes generate data that significantly differs from raw data. Mentioning these limitations would provide a more comprehensive discussion.

Annotated reviews are not available for download in order to protect the identity of reviewers who chose to remain anonymous.

---

## Round 0.2 · accepted · Accept

Thank you for submitting the article to PeerJ Computer Science. All reviewers have no further suggestions and agree to accept the article for publication.

Reviewer 2 ·

Basic reporting

NA

Experimental design

NA

Validity of the findings

NA

Additional comments

All the previous comments have been properly addressed. I have no additional comments.